# Learning Visual Representation with Synthetic Images and Topologically-defined Labels

## Abstract

We propose a scheme for neural networks to learn visual representation with synthetic images and mathematically-defined labels that capture topological information. To verify that the model acquires a different visual representation than with the usual supervised learning with manually-defined labels, we show that the models pretrained with our scheme can be finetuned for image classification tasks to achieve an improved convergence compared to those trained from scratch. Convolutional neural networks, built upon iterative local operations, are good at learning local features of the image, such as texture, whereas they tend to pay less attention to larger structures. Our method provides a simple way to encourage the model to learn global features through a specifically designed task based on topology. Furthermore, our method requires no real images nor manual labels; hence it sheds light on some of the lately concerned topics in computer vision, such as the cost and the fairness in data collection and annotation.

## 1 Introduction

Self-supervised learning (SSL) has seen a great success both practically and scientifically by providing a scheme to train neural networks without human annotated labels and by showing the similarity to humans in the learning process and the learned representation. However, there are still large gaps between SSL and human learning. In visual representation, convolutional neural networks (CNNs) are famously known to be "shortsighted" by being biased to textural information (Geirhos et al., 2019). Most image features, hand-crafted or obtained in a data-driven manner, are *local* in the sense that they are computed from patches in the image. For example, convolution, a typical and versatile form of the operation used to construct various image features, is local unless a huge kernel is used. Although local features can be aggregated by applying reduction operations such as the mean, the global information obtained in this way is limited. One way to capture global characteristics of images is to introduce a new model architecture. The attention mechanism has enabled to learn global image features in a data-driven manner as demonstrated by Vision Transformer (Dosovitskiy et al., 2021). Another way, which we concern in this paper, is to device a new training scheme, which works with virtually any model architecture with little modification. Topology is a study of shapes whose ultimate goal is to classify shapes by their topological types. Topologists have invented various *topological invariants* that can discern different shapes. For example, homology is used to classify manifolds that are locally the same Euclidean space but globally different. Our idea is to design a task of computing invariants from an input image by a neural network so that the neural network is encouraged to learn visual representation that is relevant to the topology of the image. In contrast to the popular SSL schemes, which are largely based on *learning the distribution* of real (plausible) images, our method is based on *approximating the mathematical computation* of an invariant, called persistent homology, of synthesised images.

Persistent homology (PH), one of the main tools of the emerging field of *Topological Data Analysis* (TDA), provides efficient machinery for computing global topological features of data mathematically (Adams & Moy, 2021). It has been proved to be practically useful for image processing such as classification (Dunaeva et al., 2016) and segmentation (Tanabe et al., 2021). In recent years, applications of persistent homology have expanded to many areas in science and led to new discoveries (Giunti, 2021), but in most cases, it is used just as yet another feature extractor. In view

of TDA, we proceed a step further and ask *if we can teach topology to a neural network* so that the model learns low-level image features together with the mechanism for computing high-level topological features from low-level features. Answering this question affirmatively would lead to the acquisition of more abstract and generalisable representation by the model. More precisely, our scheme is to train the model with regression of the vectorised persistent homology for synthesised images. Through this task, the model is expected to learn relevant visual representation required to approximate persistent homology. We demonstrate the validity of our scheme by experiments showing that a CNN pretrained by the proposed method can be finetuned for image classification tasks to show an improved convergence compared with one trained from scratch.

It is also worth pointing out that our scheme does not rely on real images nor labels, but uses mathematically generated images annotated with mathematically defined features. In this way, it is free from human bias which lies not only in the manual annotation but also the image themselves; photos reflect the present world and the view of the photographer. In fact, the models trained with ImageNet by SSL, even without human annotated labels, are known to be subject to bias (Steed & Caliskan, 2021).

## 2 RELATED WORK

The proposed method is built on three key ingredients, which we discuss in this section. We introduce some novel ideas to each of the three and combine them to develop our scheme.

### 2.1 SELF-SUPERVISED LEARNING ON IMAGES

Self-supervised learning tasks, which we call pretext tasks, are designed to learn image features without manually-defined labels. There are three major types of pretext tasks. The first task is to tell if given two images come from the same image or not, where variations of images are generated by applying transformations in spacial and colour domains (see Jaiswal et al. (2021) for a survey). The second task is to undone degradation, such as adding noise and masking, and reconstruct the original image. The third is similar to the second one but to perform a pair of (approximately) invertible processes, such as compression-expansion, as represented by the celebrated autoencoder (Hinton & Salakhutdinov, 2006). All of the three tasks demand the model to acquire high-level representation of the images. The main objective of these methods lies in, more or less, capturing the distribution of the training data; to be good at in-painting or compression, one has to find a low-dimensional manifold which models the training data well. We propose another type of task that put more emphasis on the computation process rather than the distribution of the data. When the computation is based on a certain mathematical structure of the data, a model will be incentivised to focus on the structure through learning the task. Our pretext task is to approximate the computation of persistent homology of the image. Persistent homology can be computed mathematically from the data and used as the label for a regression task. It is also notable that our pretext task does not rely on semantics or human perception but solely on mathematics. This allows us to use synthetic images almost meaningless to human eyes, and make the procedure completely free of real data.

### 2.2 LEARNING WITH SYNTHETIC IMAGES

Even though SSL saves the annotation costs, the preparation of training data is still a vexing problem. Publicly available datasets can be of low-quality, subject to bias, or violating usage rights and privacy. ImageNet, one of the most popular large-scale datasets, suffers from fairness issues (Yang et al., 2020), and there have been a growing interest in the fairness of machine learning Mehrabi et al. (2021). Steed & Caliskan (2021) points out that even models trained on ImageNet with SSL without using labels learn racial, gender, and intersectional biases from the way people are stereotypically portrayed on the web. No matter how much care and attentions are paid for data collection, it is impossible to be free from all kinds of these issues as long as real images are used. Using generative adversarial networks (GANs) to generate image datasets for training is a popular and successful strategy to mitigate the situation (Besnier et al., 2020), but GANs are also trained with natural images and cannot avoid above-mentioned problems. A promising approach is to use algorithmically synthesised images. Formula-driven Supervised Learning introduced in Kataoka et al. (2020) considers pretraining with synthetic images generated by a mathematical formula. The labels are assigned according to the

parameters of the image generation. Several different formuli are tested and an iterated function system, which generates fractal images, is found to be effective. A wider variety of image generation methods have been tested since then (Baradad et al., 2021; Kataoka et al., 2022). To see how synthetic images are helpful in acquiring image features is interesting also in terms of cognitive science.

In this paper, we also use synthetic images, which are not very meaningful for human eyes, and try to learn topological features from them. One technical difference from the Formula-driven Supervised Learning is how the labels are generated. Instead of fixed labels associated with the image generation model parameters, we generate labels by a mathematical formula computed directly from the images, which has advantage of being applicable to any image generation model. The labels encode topological features (persistent homology, in our paper) and the model is encouraged to learn image features that are relevant to approximate the topological features.

### 2.3 PERSISTENT HOMOLOGY FOR IMAGE ANALYSIS

Persistent homology has an unusual input and output when it is seen as a feature extractor. PH takes a series of nested topological spaces and outputs a multiset of intervals of the real numbers. We view an image as a function $f : X \to \mathbb{R}$ defined over a rectangular domain $X$, and we obtain a series of nested spaces

$$\emptyset \subset X_{t_1} \subset X_{t_2} \subset \cdots \subset X_{t_m} = X, \quad X_{t_i} = \{(x, y) \in X \mid f(x, y) \leq t_i\},$$

where $t_m = \max(f)$. Applying the homology functor $H_d(-)$ with the coefficients in the field $\mathbf{F}_2$ with two elements, we obtain the corresponding sequence of $\mathbf{F}_2$ vector spaces, and this sequence is by definition the persistent homology $PH_d(X, f)$ of the pair $(X, f)$. $PH_d(X, f)$ can be written as the direct sum of so-called the interval module having the form

$$0 \longrightarrow \cdots \longrightarrow 0 \longrightarrow \mathbf{F}_2 \longrightarrow \cdots \longrightarrow \mathbf{F}_2 \longrightarrow 0 \longrightarrow \cdots \longrightarrow 0$$

$$\| \qquad\qquad \cap \qquad\qquad \cap \qquad\qquad \cap \qquad\qquad \cap \qquad\qquad \cap$$

$$H_d(\emptyset) \to \cdots \to H_d(X_{t_{i-1}}) \to H_d(X_{t_i}) \to \cdots \to H_d(X_{t_{j-1}}) \to H_d(X_{t_j}) \to \cdots \to H_d(X).$$

We denote the summand by the interval $[t_i, t_j)$. When $d = 0$, it may happen that $j - 1 = m$, in which case we represent the summand by $[t_i, \infty)$. See Fig. 1 for an example of PH of an image. To sum up, PH takes a function $f : X \to \mathbb{R}$ and outputs a multiset of intervals. The alien output as multiset can be transformed into a fixed-length vector using vectorisation techniques (Adams et al., 2017; Bubenik, 2015; Chung & Lawson, 2021) so that it fits in the standard machine learning pipeline.

CNNs tend to be biased towards texture when trained with a classification task of natural images (Geirhos et al., 2019). In contrast, human perception relies much on the global shape of the image content. Therefore, it would be beneficial for a model to learn global topological features as well as local features. This is supported by an experiment in which combining PH with other descriptors results in an increased performance in object recognition (Li et al., 2014), showing that PH provides a shape feature that is complementary to conventional ones. There are previous studies to assimilate PH into deep learning. To deal with PH with neural networks, a parametric representation of persistence homology with learnable parameters is introduced in Hofer et al. (2017) so that a task-optimal vectorisation is obtained in a data-driven manner. Dedicated architectures of CNNs are designed in Som et al. (2020) for computing vectorised PH of time-series and point clouds in the form of the persistence image. The topological autoencoder (Moor et al., 2020) learns a latent space of a point cloud that preserves the topological structure in terms of persistent homology. Our idea is based on the fact that one has to acquire a certain high-level image representation that encodes global topological features in order to compute PH, and this makes a good pretext task for training a neural network.

## 3 METHOD

Our proposed scheme can be viewed as a type of SSL in which desired image features are learned by solving a pretext task. The pretext task does not require any natural images, but synthesised images (Sec. 3.1) are used. The model learns the input-output relation (regression) for the image and the corresponding image feature vector computed through persistent homology (Sec. 3.2). The model is

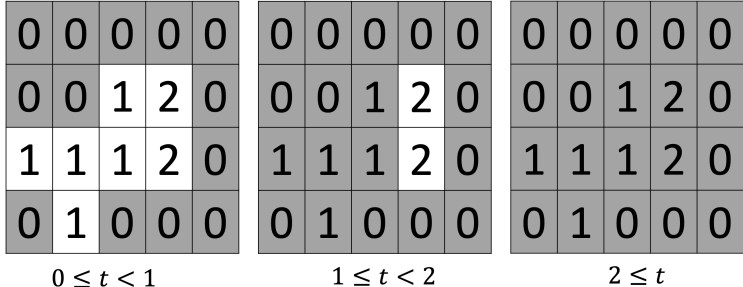

Figure 1: The figure shows the sublevel sets of an image for different ranges of $t$. For an image, persistent homology consists of two components $PH_0(X, f)$ and $PH_1(X, f)$, which respectively records the transition of islands and holes in the sublevel sets under different threshold. In this example, $PH_0(X, f) = \{[0, 1), [0, \infty)\}$, where the interval $[0, 1)$ corresponds to the connected component consisting of the single pixel at the bottom-left corner in the left-most figure, which is merged to the other connected component in the central figure. Since this feature exists for $t \in [0, 1)$, it is represented by the interval $[0, 1)$, and said to have the birth time 0, death time 1, and life time $1 (= 1 - 0)$. The other component that exists for $t \geq 0$ is represented by $[0, \infty)$, which has the infinite life time. Similarly, $PH_1(X, f) = \{[1, 2)\}$ whose element represents the hole surrounding the two pixels with the value 2 in the central figure. This hole disappears in the right-most figure, and so, it is represented by $[1, 2)$. Persistent homology of an image records the topological features, such as islands and holes, with the threshold in which the features emerge and disappear. The distinctive idea of *persistence* is to trace the emergence and disappearance rather than computing features at each threshold separately.

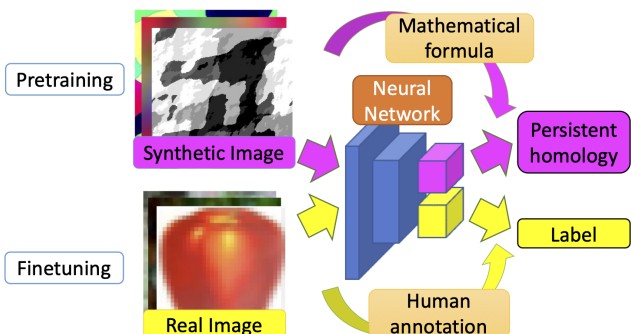

Figure 2: The overall structure of the proposed method.

expected to learn relevant image features which are required for approximating PH. Intuitively, the procedure is described as follows. The teacher knows topology and gives problems to the student with the answer computed mathematically. The student tries to guess how the teacher's answer is computed and search for clues in the image. So it is not guaranteed that the student learns how to compute PH in the way it is defined mathematically, but the student collects image features that are helpful to approximate PH, and those features are necessarily global and topological, as so is PH. Similarly, any algorithmically computable vector associated with the image can be used. However, we particularly choose PH as our image feature since (1) it captures global topological characteristics of the image and is known to have discriminative power (Adams et al., 2017), (2) it has guaranteed stability (including robustness against the change in the pixel values and the invariance against isometric geometric transformations (Cohen-Steiner et al., 2005; Skraba & Turner, 2021)), and (3) it is defined for any image and is efficiently computed (Kaji et al., 2020). We elaborate on these points below.

(1) The image features obtained by PH are topological and global, whereas the features acquired by CNNs through supervised learning with ImageNet are often biased towards textures. We expect that learning abstract topological features first in pretraining and local features such as texture in

finetuning will have complementary effects.

(2) Learning invariant features is one of the main principles in visual representation learning (Misra & Maaten, 2020). PH gives the same labelling for images that are rotated and reflected, and similar labelling for images with certain types of pixel values alteration, which is guaranteed by the stability theorems. The regression of PH encourages the model to learn those invariances. Furthermore, when a certain transformation of the image which changes its PH is applied, the model is asked to learn the *change in the label*; persistent homology is *functorial*. That is, the transformation in the input is systematically reflected by the transformation in the output, and the model is asked to learn this higher relation as well when trained with data augmentation such as by affine transformations.

(3) For randomly generated images, we cannot rely on semantics of the images that is meaningful for human eyes to design the pretext task. We do not have this problem with PH, which is mathematically defined. Moreover, computation of PH is cheap (Appendix A.1) so that it does not increase overall pretraining time substantially.

We evaluate our scheme[1] by finetuning the model pretrained by our scheme for an image classification task with some popular datasets (Figure 2 depicts a schema). We finetune the model with the training dataset and the performance in terms of the convergence of the classification accuracy for the validation dataset is measured. At the beginning of finetuning, we replace the fully-connected layer with a randomly initialised one. We remark that the topological features encoded by persistent homology alone are insufficient for image classification tasks, and they are meant to complement local features such as texture. For this reason, the weights of all layers are updated during finetuning.

The experiments are conducted with our codes publicly available under the MIT license at `https://XXX` (the URL is removed temporarily for the double-blind review. The codes are found in supplemental materials.).

## 3.1 IMAGE GENERATION

We use the following frequency-based random image generation method to create a dataset for pretraining. We choose this image generation model mainly because it is computationally inexpensive and produces images with various frequency profiles (see also Appendix A.5). In particular, the resulting images (see Fig. 3) contain patterns at different scales, and hence, have rich topological information.

1. Create an array $h$ of dimension $256 \times 256$, where the values are drawn independently from the uniform distribution on $[0, 1]$.

2. For a frequency parameter $\beta$ drawn from the uniform distribution on $[1, 2]$, set

$$g(x, y) = \mathrm{Re}\left( \mathrm{iFFT}\left( \frac{\mathrm{FFT}(h)(x, y)}{((x + 1)^2 + (y + 1)^2)^\beta} \right) \right), \tag{1}$$

where FFT is the 2D discrete Fourier transform and iFFT is its inverse, and $\mathrm{Re}$ denotes the real part of a complex number.

3. With a probability of $p$, which we set to 0.5, binarise $g$ by Otsu's thresholding Otsu (1979).

4. Repeat the process three times independently to create a colour image with RGB channels.

5. Convert the image into greyscale with a probability $q = 0.5$.

The random parameter $\beta$ controls how fast the high-frequency components decay. The effect of the choices for the hyper-parameters $p, q$ and the range of $\beta$ does not seem to be large and we haven't done a comprehensive search. We have only checked that setting $p = 0.5$ and $q = 0.5$ is slightly better than $p = 0, 1$ or $q = 0, 1$.

## 3.2 PH LABEL COMPUTATION

The images labels for the pretext regression task are computed using persistent homology. There are various ways to utilise PH to encode topological information of images (see Turkeš et al. (2021);

---

[1]Some more experiments which corroborate or complement this evaluation experiment are given in Appendix A.2 and Appendix A.4.

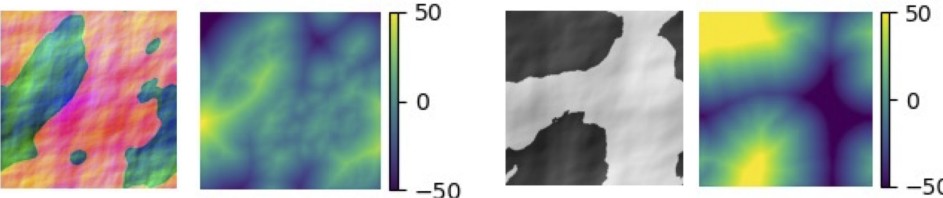

Figure 3: Two examples of synthesised images and their signed distance functions. Persistent homology is computed for the sequence of the sublevel sets for the signed distance function.

Garin & Tauzin (2019) for a survey). Our choice aims at extracting shapes contained in the image by their contours and capturing its topological features such as connected components and holes, together with their scale. To this end, a nested sequence of spaces, which is the input for PH, is built from the image by the following procedure: (1) Convert the image into greyscale (2) Binarise it using Otsu's thresholding (Otsu, 1979) (3) Compute the signed distance function (see Figure 3)

$$\phi(x,y) = \begin{cases} -\min_{(x',y'):\text{background}} |(x,y) - (x',y')| \\ \qquad \text{if } (x,y) \text{ is foreground} \\ \min_{(x',y'):\text{foreground}} |(x,y) - (x',y')| \\ \qquad \text{if } (x,y) \text{ is background.} \end{cases} \tag{2}$$

(4) Finally, obtain the sequence of the sublevel sets $X_t = \{(x,y) \in X \mid \phi(x,y) \leq t\}$, where $X$ is the domain of the original image; that is, the rectangular grid of the size of the image. This procedure gives a bounded nested sequence satisfying $X_s \subset X_t$ when $s < t$ and $X_{-D} = \emptyset, X_D = X$, where $D$ is the length of the diagonal of the image. The PH (with the coefficients in the field $\mathbf{F}_2$) of this sequence is computed by the software called Cubical Ripser (Kaji et al., 2020). An important remark is that instead of considering the sublevel sets with respect to the original (greyscaled) pixel values, we consider the sublevel sets of the signed distance function. In this way, the resulting PH captures the metric-aware structure of the original image; the scale of topological features are encoded as life time in persistent homology.

The degree 0 and 1 parts of persistence homology are vectorised separately and concatenated into a single vector, which serves as the label for the image. We test four different vectorisation techniques; the persistence image (Adams et al., 2017), the persistence landscape (Bubenik, 2015), the Betti number curve (Chung & Lawson, 2021), and the birth-life histogram. The last one is simply the histograms of the birth time (the left end of the interval) and the life time (the length of the interval) of the persistence homology for each degree. All the vectorisation techniques have hyper-parameters for the "resolution" of the output, which are determined from the specified output dimension in our experiments.[2] In most of the experiments, the output dimension is fixed to 200. We allocate the same dimension for the degree 0 and 1 persistence homology so that we obtain a pair of 100-dimensional vectors that are concatenated to form a 200-dimensional output vector. To reduce the dynamic range and suppress overflow, the square root is taken for each coordinate of the vector. Although we may elaborate on the hyper-parameter tuning for the PH vectorisation, Tab. 1 shows that even the choice of the vectorisation techniques does not have a large impact. Therefore, we limit ourselves to an essential set of experiments to find out the nature of the scheme as detailed in Sec. 3.3.

### 3.3 EVALUATION RESULT

We evaluate the effect of the pretraining by looking at the convergence of the accuracy in finetuning for classification tasks. Along with the popular ImageNet-1k (IMN-1k) (Deng et al., 2009) and CIFAR100 (C100) (Krizhevsky, 2009) datasets consisting of natural images, we use the animal dataset (ANM) (Bai et al., 2009), which is of relatively small size and consists of 2,000 binary images of animal contours of various size that are labelled with 20 classes. Since no texture information is present in the animal dataset, it is suitable to verify the effect of topology learning, and is used in

---

[2]The only non-canonical choices among the standard set of hyper-parameters of the vectorisation techniques are the number of *landscape functions* for the persistence landscape and the sigma of the Gaussian kernel used in the persistence image, which we set to 2 and 1.0 respectively.

Hofer et al. (2017) for evaluation of their method of incorporating PH into deep learning. We split the animal dataset into two sets with 1,600 training images (80 images per class) and 400 validation images.

The hyper-parameters for learning are fixed in a standard manner as follows. ResNet50 network architecture is used. Input images are resized to $256 \times 256$ and then cropped randomly down to $224 \times 224$. Random horizontal flipping is applied. We vary the number of epochs according to the dataset; both in pretraining and finetuning, 90 epochs for the CIFAR100 dataset, 90 epochs for synthesised dataset, 300 epochs for the animal dataset, and 45 epochs for ImageNet. The stochastic gradient descent with a momentum of 0.9 and weight decay of $10^{-4}$ is used as the optimiser. The initial learning rate is set to 0.1 for pretraining (0.01 for finetuning) and multiplied by 0.1 twice at the 1/3 and the 2/3 of the total training epochs. The batch size is set to 128.

Table 1 shows the classification accuracy of the validation dataset after each pretrained model is finetuned with the training dataset using its labels. The finetuning phase is exactly the same for all pretrained models. **Scratch** is without any pretraining. **Label** is pretrained with the training dataset using its labels.[3] That is, the model is trained twice with the training dataset but with different learning rates. The four PH-based models are pretrained with the synthetic dataset with 400,000 images described in Sec. 3.2 with different PH vectorisation targets (**PH-PI** with the persistence image, **PH-LS** with the persistence landscape, **PH-BC** with the Betti number curve, and **PH-HS** with the birth-life histogram). For comparison, a model pretrained by a popular SSL scheme, **MoCo-v2** (Chen et al., 2020), with the same synthetic dataset with 400,000 images is shown. As a comparison target for pretraining with synthesised images, the results of the FractalDB-pretrained models (**FDB-1k** consists of 1,000,000 images with 1,000 classes and **FDB-10k** consists of 10,000,000 images with 10,000 classes) that are publicly available[4] are shown. The values in Tab. 1 differ slightly from the ones reported in Kataoka et al. (2020), possibly due to minor differences in the experimental settings. We also list the accuracy of the ImageNet pretrained model (**IMN**) that is provided as a part of PyTorch's torchvision library (version 0.10.1). Although publicly available pretrained weights are used for **FDB-1k** and **IMN**, the hours taken for pretraining with **FDB-1k** and **IMN** are measured on the same system used for pretraining other models. For **FDB-10k**, the time of **FDB-1k** is extrapolated by simply multiplied by ten.

Since there are only 400 validation images for the animal dataset, the accuracy values fluctuate. The values listed in Tabs. 1 and 2 are the mean for the last epochs from 290 to 300. The 90 percentile falls within $\pm 0.6$ of the mean. Moreover, to see the stability of the results, we have performed ten trials of pretraining and finetuning with different random seeds for the entry of **PH-PI** for C100 in Tab. 1, and observed that the 90 percentile falls within $\pm 0.3$ of the mean.

The PH-pretrained models show better convergence than **Scratch** and **Label**, and comparable convergence to **FDB-10k** even though the PH models are trained with a much smaller number of 400,000 images and about 70 times less computation time. All PH-pretrained models show more or less similar convergence regardless of the used PH vectorisation techniques. Among them, **PH-BC** performs the worst, and this can be attributed to the fact that only **PH-BC** does not care about the "persistence"; **PH-BC** can be computed just as the sequence of the Betti numbers of the sublevel set $X_t$ one $t$ at a time without considering the relation among different $t$'s, while the other three, **PH-PI**, **PH-LS**, and **PH-HS**, capture the relation. This suggests that not only homology but also its persistence structure helps learn visual representation.

Figure 4 shows the transition of the accuracy in the finetuning of the models selected from Table 1 for the CIFAR100 dataset. A similar figure for the ImageNet dataset is given in Figure 9. The ImageNet-pretrained model **IMN** surpasses all other models in both training and validation. The convergence behaviour of **PH-PI** and **FDB-10k** are quite similar, sitting between **IMN** and **MoCo-v2**.

To see the effect of various parameters and the nature of our method, we conduct a few more experiments. We fix the vectorisation method to the persistence image. Unless otherwise stated, the output dimension is fixed to 200 and the size of the synthetic image dataset is fixed to 200,000.

---

[3]The deteriorated accuracy of Label for Animal compared to Scratch is likely due to over-fitting. We observe the validation loss converges quickly although to a higher value.

[4]https://github.com/hirokatsukataoka16/FractalDB-Pretrained-ResNet-PyTorch

Table 1: Classification accuracy of models pretrained with various methods. For FDB-1k and FDB-10k, the pretrained weight file made publicly available by the authors of Kataoka et al. (2020) is used. For IMN, the weight file that comes with the torchvision library is used. The hours taken for pretraining is measured on a PC with a single NVIDIA RTX 3090 and an Intel Core i9-10850K.

| | Scratch | Label | PH-PI | PH-LS | PH-BC | PH-HS | MoCo-v2 | FDB-1k | FDB-10k | IMN |
|---|---|---|---|---|---|---|---|---|---|---|
| C100 | 69.6 | 70.3 | 78.4 | 78.1 | 76.6 | 77.9 | 71.6 | 75.3 | 78.1 | 85.0 |
| ANM | 80.7 | 80.1 | 91.0 | 90.1 | 89.1 | 90.6 | 85.0 | 84.6 | 85.2 | 93.3 |
| IMN-1k | 70.8 | NA | 72.9 | 72.0 | 71.4 | 72.8 | 71.0 | 71.8 | 72.4 | 75.3 |
| time | 0h | 3h | 22h | 22h | 22h | 21h | 38h | 144h | 1442h | 74h |

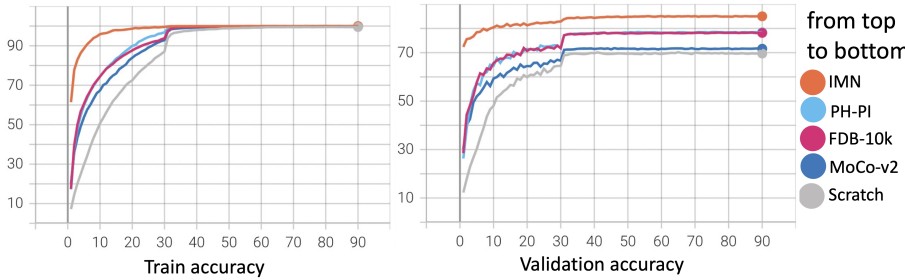

Figure 4: Transition of the training and validation accuracy in the finetuning of the CIFAR100 dataset.

The impact of the choice of the vectorisation dimension of PH is assessed by varying the value among $\{100, 200, 400, 800\}$. As we observe in Tab. 2 (left), the choice affects the performance and the optimal value, in this case, is about 200. However, we guess the optimal value depends on the complexity and the dimension of the synthesised images as the vectorisation dimension controls the resolution of the discretisation of PH.

The impact of the size of the pretraining dataset is assessed by varying the size of the synthesised dataset among $\{50,000, 200,000, 400,000, 800,000\}$. As is observed in Tab. 2 (middle), the performance increases as the size gets larger. At the size of 50,000, which is the same as that of the CIFAR100 training dataset, the **PH-PI** model performs better than **Label** in Tab. 1, which is pretrained with the same number of images and iterations with the same learning rates.

Table 2: (Left) Classification accuracy of PH-PI models pretrained with different dimensions of the PH vectorisation (the dimension of the vector of the regression target). (Middle) Classification accuracy of PH-PI models pretrained with the synthesised dataset of different sizes. The values for 400k are reproduced from Tab. 1. (Right) Classification accuracy of PH-PI models pretrained with with the CIFAR100 (PH-C) and the animal (PH-A) datasets. The values for Scratch are reproduced from Tab. 1.

| | 100 | 200 | 400 | 800 | 50k | 200k | 400k | 800k | Scratch | PH-C | PH-A |
|---|---|---|---|---|---|---|---|---|---|---|---|
| C100 | 76.8 | 77.7 | 77.4 | 75.0 | 76.1 | 77.7 | 78.4 | 78.8 | 69.6 | 75.3 | 72.4 |
| ANM | 87.4 | 89.9 | 90.5 | 87.5 | 88.6 | 89.9 | 91.0 | 91.6 | 80.7 | 86.5 | 83.2 |

Our pretraining scheme also works with real images in place of synthetic ones. In the next experiment, we pretrain the model with the training split of the CIFAR100 dataset (and the animal dataset) using the PH-regression task, and then finetune the model using the class label of the dataset. The result is shown in Tab. 2 (right). In this example, the setting of **Label** and **PH-C** (**PH-A**, respectively) for the CIFAR100 dataset (the animal dataset, respectively) is the same except for the label used for the pretraining; **Label** uses the class label while **PH-C** (**PH-A**, respectively) uses the label computed with the persistence image. Pretraining with PH improves the performance, indicating the benefit of learning not only from the class labels but also with topology-based labels. Comparing with the entry for 50k in Tab. 2 (middle), we see our synthetic dataset offers slightly better quality for learning

than the CIFAR100 dataset, which consists of the same number of 50k natural images. This could be attributed to the design of the image generation model that produces patterns at various scales. In the case of the animal dataset, pretraining with PH does not lead to a large performance gain. This may be due to the small number of training images (1,600) and the variety is quite limited. The observation agrees with the result in Tab. 2 (middle) that indicates the necessity of a certain amount of data for learning topological features.

## 4 LIMITATIONS

We have seen that CNNs can learn useful image features from a regression task of persistent homology. However, (vectorised) PH alone is not a strong image feature for a classification task although it captures different characteristics of images than conventional techniques. We believe adding topology is helpful but does not replace textural information that CNNs naturally learns from natural image classification. Our proposed scheme forces the model to focus on topology during pretraining; the PH label is computed from binarised and signed distance transformed images from which textural information is stripped off. The PH label encodes topological features with their scale. The model learns other local features only during finetuning, where topology is not paid a particular attention. Although this splitting of topology learning from conventional learning makes the analysis more controlled, it would not be the optimal strategy to learn both local and global image features if the primary goal is a better performance in practical applications.

Another limitation, which is related to the explainability just as in most deep-learning schemes, is that we are not really sure what features are learned in our method although we have designed it for learning global topological features of images. We have observed that both the training and the validation losses converge during pretraining the PH-based models, but the latter stays about twice higher than the former, indicating that the models learn to *approximate* PH but fails to learn to *compute* PH in a fully generalisable (that is, mathematical) manner. Nonetheless, we have observed that the visual representation obtained by the model pretrained by PH is quite different from those pretrained by FractalDB and ImageNet (see Appendix A.2). From a practical viewpoint, this means the model would have strengths and weaknesses depending on the downstream tasks. Detailed and controlled experiments including more than simple classification tasks should be performed to verify the usefulness of the learned features across different tasks.

## 5 CONCLUSION

We proposed a scheme for learning visual representations that were relevant to the topological features of images through a regression task for a mathematical function defined by persistent homology. Our method can be applied to any image dataset and virtually any neural network architecture with a small cost. Our experiment with a synthetic image dataset suggested that even images that were little meaningful for human eyes could be used for learning certain visual representation. Our main contributions are two fold: (I) We proposed a pretext task that was mathematically defined and applicable to synthetic images without labels. (II) We experimentally showed that a convolutional neural network pretrained with the pretext task acquired visual representation that was different from those obtained by the usual supervised learning.

We would like to conclude the paper with a few possible future directions to investigate. (a) Since global topological features are generally robust compared to local ones, learning them could add resistance against adversarial attacks. (b) Regressing other mathematical invariants could be used to pretrain neural networks to equip them with understanding of geometric, topological, and algebraic structures of the image. This would provide a generic method to incorporate various knowledge established by mathematicians over past centuries to deep learning research. In terms of learning with synthetic images, regressing the random parameters used in the image generation models such as in (Kataoka et al., 2020; Baradad et al., 2021) would also be interesting. (c) A theoretical question related to this work is on the relation between the neural network architecture and the learning capacity. It is known that universal-approximation-theorem-type statements have some constraints that relate topology of the target function to be learned and the width of neural networks (Johnson, 2019). It would be interesting if we could compare the way in which a CNN learns to approximate PH and the rigorous mathematical definition of PH.

## REPRODUCIBILITY STATEMENT

The experiments are conducted with our codes included as supplemental materials. We will release them at github under the MIT license. Robustness of the numbers in the tables and figures are discussed in Section 3.3.

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

## A  APPENDIX

### A.1  COMPUTATIONAL COST OF PH LABELLING

The worst-case complexity of the standard matrix reduction algorithm for computing persistent homology is $K^3$, where $K$ is the number of columns, which is linear with respect to the number of pixels in the cubical complex. The average time of computation for persistent homology and its persistence image as in the PH-PI pretraining for a synthesised image described in Sec. 3.1 in different sizes is shown in Tab. 3. The increase rate looks almost linear with respect to the number of pixels in this particular set of images.

Table 3: Average computation time for PH and persistence image

| image size | $256 \times 256$ | $512 \times 512$ | $1024 \times 1024$ | $2048 \times 2048$ |
|---|---|---|---|---|
| computation time in ms | 22.62 | 89.85 | 436.93 | 2181.13 |

### A.2  WHAT IS REALLY LEARNED?

Understanding and explaining what the model is learning is a difficult problem, which forms one of the central topics in machine learning research. Here, we show some experiments that indicate the difference between the models trained in our scheme and in the classical supervised manner with real

images and manual labels. First, we observe a distinction in the weights of the first few convolution layers among different models. Figure 5 shows that the model learns different low-level features through our scheme.

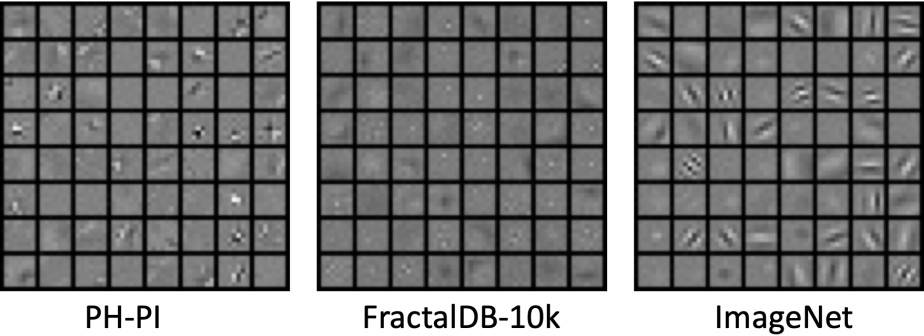

PH-PI                    FractalDB-10k                    ImageNet

Figure 5: Visualisation of the filters of the first convolutional layer of PH-PI, FractalDB-10k, and ImageNet pretrained models.

To understand this phenomenon, we conducted an experiment "the other way around" to see if the visual representation acquired through supervised learning with real images is sufficient to approximate persistent homology. Specifically, we replaced the fully-connected layer of the ImageNet pretrained model with a randomly initialised one. Then, the model was finetuned for the PH-regression task as in the PH-PI pretraining while the weights of all but fully-connected layers were frozen (we call this model **IMG_fr**). The model was compared with the one trained from scratch (by updating all layers), which was precisely in the same manner as in the PH-PI pertaining (we call this model **Scratch**). Figure 6 shows that the validation loss, as well as the training loss, converges for **Scratch**, and hence, the model was learning to approximate PH to some extent. However, we observe that the convergence of **IMN_fr** is much worse than **Scratch**, indicating that the image features learned by a supervised training with ImageNet is not sufficient for approximating PH. That is, not only fully-connected layers but also convolutional layers need to be trained for the computation of PH. This explains what we have observed in Fig. 5; the convolutional layers of **PH-PI** learn features specific to PH computation, which are different from those features learned with a supervised image classification task with ImageNet. This behaviour sharply contrasts with the fact that the ImageNet pretrained model **IMN** can be finetuned for various natural image classification tasks by updating only the weights of fully-connected layers. The convolutional layers of **IMN** have obtained image features sufficient for general classification tasks of natural images, which are insufficient for approximating PH.

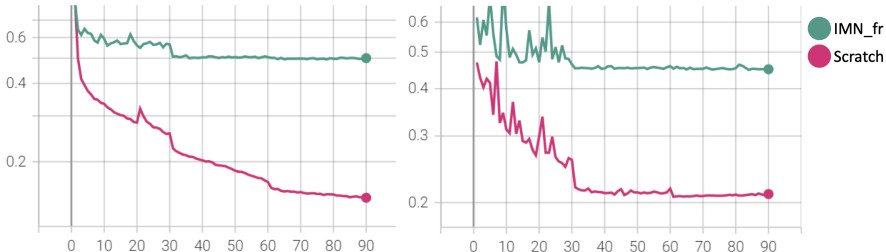

Figure 6: Transition of the loss for the PH-regression task for the ImageNet pretrained model with the weights of all but fully-connected layers frozen (**IMN_fr**) and a model with randomly initialised weights (**Scratch**).

To investigate this direction further, we performed another experiment of the PH-regression with a different set of synthesised images with simple random blobs which had virtually no texture information (Fig. 7 (Top)). We finetuned the ImageNet pretrained (**IMN_fr**) and the PH-PI pretrained

(**PH-PI_fr**) models while the weights of all but fully-connected layers were frozen. The transition of the loss values is shown in Fig. 8. The validation loss of **PH-PI_fr** converges to a lower value than that of **IMN_fr**. This shows the visual representation learned by the convolutional layers of **PH-PI** is useful to approximate the PH of a different type of image than was used for pretraining. On the contrary, the features learned by pretraining with ImageNet are not sufficient to approximate PH, even for very simple images. Figure 8 also shows the transition of the loss values for **IMN** and **PH-PI**, where all layers including convolutional ones were updated during finetuning. Both models had better convergence than the ones updating only the fully-connected layers, but **PH-PI** had lower loss values than **IMN** indicating that updating all layers did not completely override the learned features during pretraining.

The visualisation of Grad-CAM++ (Chattopadhay et al., 2018) (with the target of the last convolution layer) of the two models, **PH-PI_fr** and **IMN_fr**, is shown in the middle and bottom rows of Fig. 7. Although it is hard to interpret, we may be able to say, **PH-PI_fr** (Bottom) identifies a large circle as a single shape whereas **IMN_fr** (Middle) has more focus on edges and finds a large circle as a collection of two or three shapes.

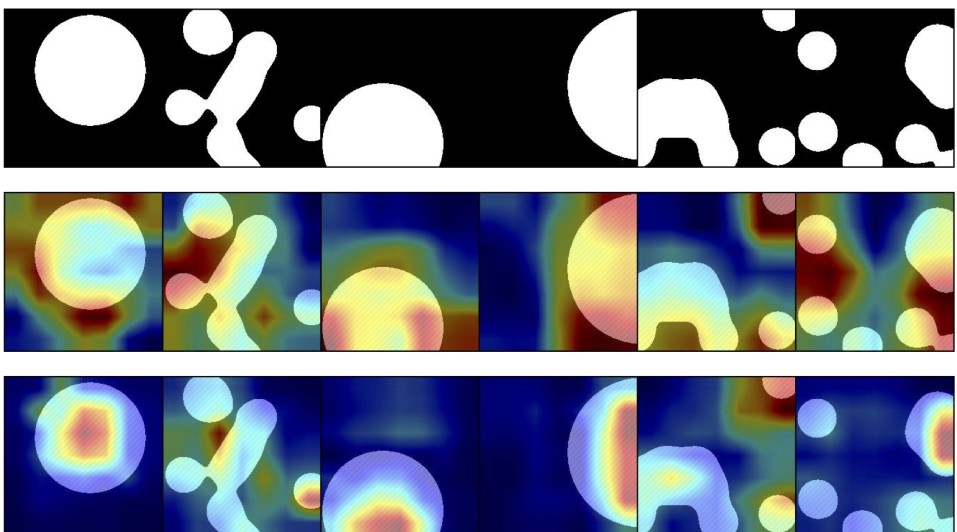

Figure 7: (Top) randomly synthesised blob images (Middle) Grad-CAM++ visualisation of **IMN_fr**, and (Bottom) **PH-PI_fr**

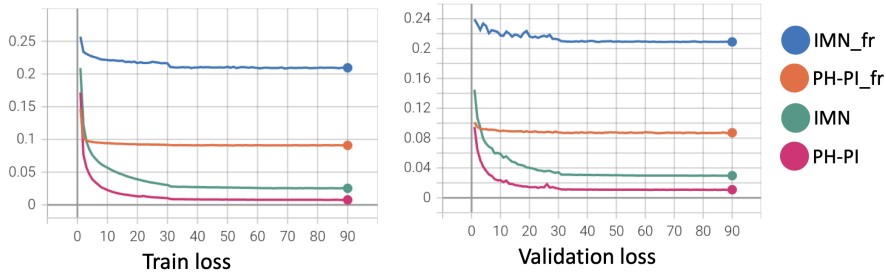

Figure 8: Transition of loss for PH-regression task with the random blob images. The ImageNet pretrained model (**IMN**) and the PH-PI model (**PH-PI**) pretrained with the synthetic dataset described in Section 3.1 are finetuned to compute the persistent homology (vectorised by persistence images) of another set of synthetic images (Fig. 7(Top)). Corresponding models with the weights of all but fully-connected layers frozen during finetuning are named **IMN_fr** and **PH-PI_fr**, respectively. Note that the two sets of randomly generated images (Fig. 7(Top) for finetuning and Fig. 3 for pretraining **PH-PI**) are very different.

### A.3 Transitions of statistics during finetuning

The transition of the accuracy during the finetuning of the models selected from Table 1 for the ImageNet-1k dataset is given in Fig. 4. We observe a similar tendency to Fig. 4 for the CIFAR100 dataset. The ImageNet-pretrained model **IMN** surpasses all other models in both training and validation. The convergence behaviour of **PH-PI** and **FDB-10k** are quite similar, sitting between **IMN** and **Scratch**.

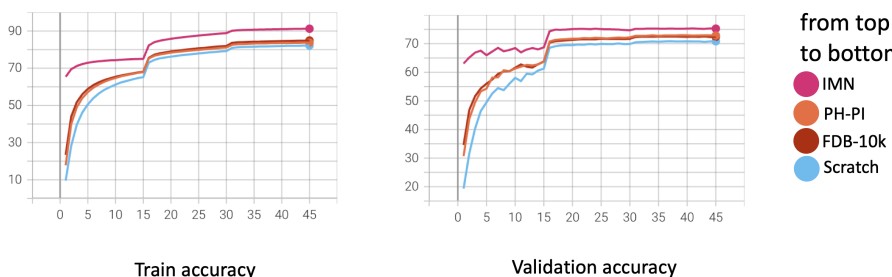

Figure 9: Transition of the training and validation accuracy during the finetuning of the selected models with ResNet50 for the ImageNet-1k dataset.

To see if the proposed scheme is effective for a more powerful CNN architecture, we performed an experiment with ResNet101. The experiment configuration was the same as in Table 1 except for the model architecture. Figure 10 shows the transition of the accuracy during the finetuning of the selected models with ResNet101 for the CIFAR100 dataset. Note that due to the large computational time required for pretraining **FDB-10k**, we could not include **FDB-10k** in the figure.

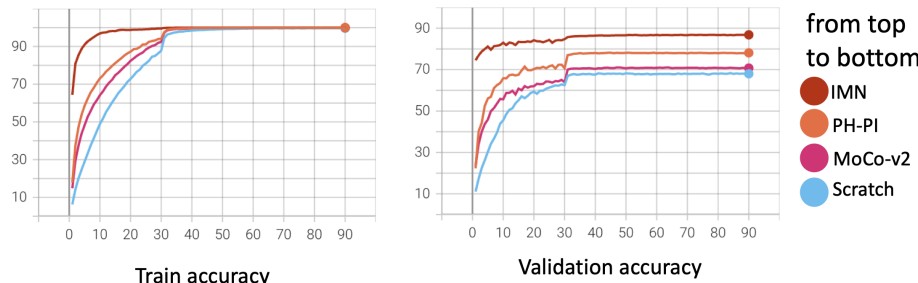

Figure 10: Transition of the training and validation accuracy during the finetuning of the selected models with ResNet101 for the CIFAR100 dataset.

As an example of another downstream task than classification, we performed object detection with Faster R-CNN (Ren et al., 2015) for the PASCAL VOC 2007 dataset. We used the trainval split for training and the test split for validation. We tested five ResNet50 models in Table 1, **Scratch**, **PH-PI**, **MoCo-v2**, **FDB-10k**, and **IMN** as the backbone of Faster R-CNN. Figure 11 shows the transition of mAP for the first 20 epochs of training and validation. The mAP observed in our experiment was very low for all SSL models. It seems that the backbone network should be pretrained in a supervised manner with a large dataset as in the original paper of Faster R-CNN. Therefore, it may not be meaningful to do any comparison, but we see that mAP increased faster with **PH-PI** than the other SSL models and training from scratch.

### A.4 Class-wise inspection on the CIFAR100 classification

Here, we investigate what kind of images are better classified with our scheme. We looked at the predictions for the CIFAR100 validation dataset (100 classes × 100 images) of the three models, **Label**, **PH-PI**, and **IMN** selected from Table 1, after finetuning. We computed the $F_1$-score for each

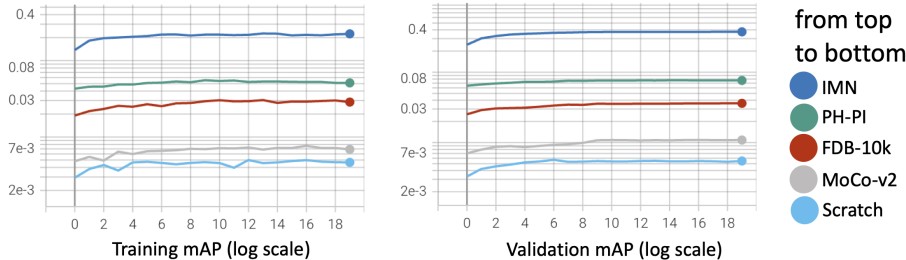

Figure 11: Transition of mAP during the finetuning of the selected models with Faster R-CNN with ResNet50 backbone for the PASCAL VOC 2007 test dataset.

Table 4: $F_1$-scores of the predictions of selected models from Table 1 for the validation dataset of C100. The list is sorted in the increasing order of the difference in the $F_1$-score between **PH-PI** and **IMN** and we show the first three and the last three classes.

| class | Label | PH-PI | IMN | (PH-PI)-(IMN) | (PH-PI)-(Label) |
|-------|-------|-------|-----|---------------|-----------------|
| bear  | 0.524 | 0.653 | 0.854 | -0.201 | 0.129 |
| snake | 0.617 | 0.670 | 0.831 | -0.161 | 0.053 |
| shrew | 0.545 | 0.573 | 0.725 | -0.152 | 0.028 |
| . . . |       |       |     |        |       |
| plate | 0.710 | 0.777 | 0.772 | 0.006 | 0.068 |
| road  | 0.906 | 0.927 | 0.916 | 0.011 | 0.020 |
| girl  | 0.472 | 0.653 | 0.623 | 0.030 | 0.181 |

of the 100 classes and sorted the result in terms of ($F_1$-score of **PH-PI**) - ($F_1$-score of **IMN**). Having a positive value for ($F_1$-score of **PH-PI**) - ($F_1$-score of **IMN**) means that **PH-PI** performed better than **IMN** for the class. The result is shown in Table 4. Although it is not shown on the list, we note that **PH-PI** had improved $F_1$-scores than **Label** for all 100 classes.

Figure 12 shows sample images from the six classes that appear in Table 4. We selected three images from each class in the following manner. For the classes of bear, snake, and shrew, with which **IMN** performed better then **PH-PI**, we selected those images where the prediction of **PH-PI** was incorrect and that of **IMN** was correct. Then, we chose three images with the lowest ranking of the true label in the prediction of **PH-PI**. For the classes of plate, road, and girl, with which **PH-PI** performed better than **IMN**, we did the same by swapping the role of **PH-PI** and **IMN**. The caption below each sample image is the prediction of the incorrect model. How the models make incorrect predictions is interesting and it seems to agree with our claim that our model put more emphasis on the shape than the texture. Especially the plate class provides an illustrative example; **PH-PI** focuses on the round shape, while **IMN** looks at the texture of the plate. We give a few more interpretations of the result: The snake in the second row can be mistaken as a bottle if we look at its silhouette without its distinctive texture. In the first row, the shrew's pose with the forefoot resembles that of a lizard. The girl in the third row wears colourful striped clothes but cannot be mistaken as a lizard by a human. All images of road consist mainly of large and simple geometric shapes, which are easily characterised by persistent homology.

## A.5 IMAGE SYNTHESIS METHOD

There are many different ways to synthesise images, such as various noise generation methods. We have not conducted a thorough test on dataset creation since the focus in the design of our pretext task is not on the distribution of the data. The key is in the process of approximating a certain computation (in our case, persistent homology) based on mathematical structure. We list some reasons why we chose the particular image synthesis method. We can see from Eq. (1): (i) It is rotation invariant, except for the fact that the image is rectangular. (ii) It is scale (image size) independent. (iii) It produces various frequency profiles: The lifetime profile of the persistent homology is a very important factor of the topology of the image. With the signed distance transform, the total persistence (the sum of lifetime of all cycles) is bounded so that we can have either many short lifetime cycles

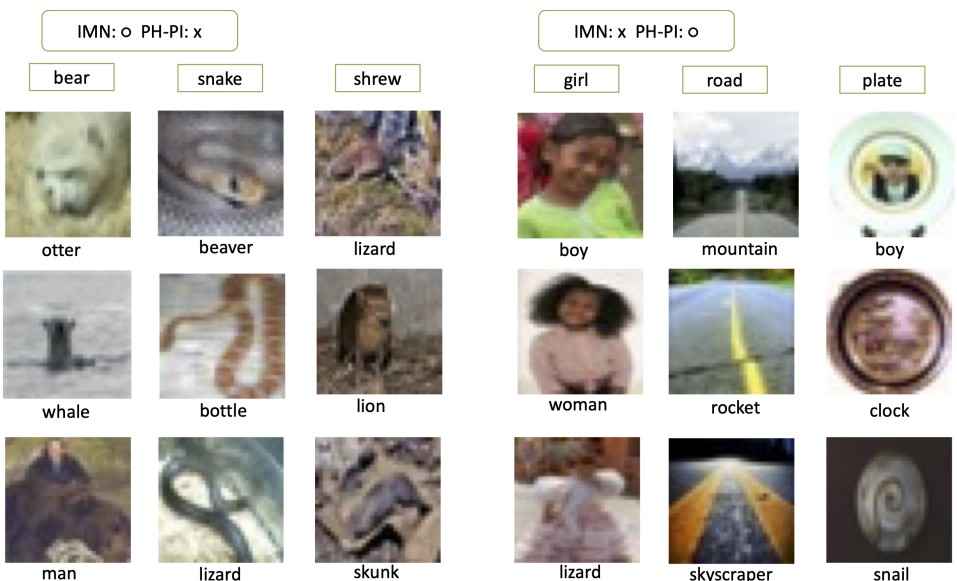

Figure 12: Sample images of the classes in Table 4. The label of each image indicates the (incorrect) prediction by **PH-PI** (for the classes bear, snake, and shrew) and by **IMN** (for the classes girl, road, and plate).

or a small number of long lifetime cycles. In natural images, short lifetime cycles arise mainly from noise or texture, which are local. The denominator of Eq. (1) discourages the resulting image from having many high-frequency components so that there are more long lifetime cycles, which represent global topology. (iv) It is computationally cheap. We emphasise that the fractal model used in Formula-driven Supervised Learning (Kataoka et al., 2020) requires tremendous computational power and their datasets are prepared on a computing cluster. Creation of our dataset with 400,000 images takes less than twenty minutes on a personal computer, and is generated on the fly during the first epoch of training.

