# OpenReview forum: "Learning Visual Representation with Synthetic Images and Topologically-defined Labels"
_ICLR.cc/2023/Conference — Submitted to ICLR 2023_

### Official Review · Reviewer_tB82 · 2022-10-24

**Confidence:** 4
**Correctness:** 2
**Technical Novelty And Significance:** 2
**Empirical Novelty And Significance:** 2
**Recommendation:** 5

**Clarity, Quality, Novelty And Reproducibility:**

Clarity: This paper is well written and easy to follow.

Quality: This paper contributes some new ideas but the empirical evaluation is limited.

Novelty: The proposed idea is somehow interesting.

Reproducibility: I am optimistic to reproduce the main results of this paper as the paper provides lots of details.

**Strength And Weaknesses:**

**Strengths**:

1. The idea of guiding the neural networks to learn global structures with non-local features, like persistent homology is novel and interesting.

2. Pre-trained with synthetic images generated by specific formulas seems to prevent the human bias existing in the annotated datasets and ensure fairness to some extent.

3. The paper is clearly structured and well written.

**Weaknesses**:

1. The authors only provide empirical results, without giving theoretical analysis of the proposed pre-training scheme.

2. The authors mention the weights of all layers are updated during the phase of fine tuning. However, it is unclear that the neural network is still able to preserve the learned global topological features if we update all layers’ parameters during the phase of fine tuning.

3. The experiments are all conducted on small datasets of natural image classification tasks, which is not sufficient to indicate that the proposed scheme can be used in other larger datasets or different downstream tasks.

**Questions**:

1. It may be helpful to conduct experiments on how the learned global features change between the pre-training phase and the fine-tuning phase.

2. Using hand-crafted features to guide the training of neural networks seems alright. However, how to prove that your hand-crafted features actually reflect the global information of the training dataset? I would expect more theoretical analysis to support the main claim of this paper.

3. It seems that the models are only tested on rather small datasets within only the natural image classification tasks. Is it possible to conduct experiments on larger datasets on other scenarios, like segmentation or tracking, to see the performance?




**Summary Of The Paper:**

This paper proposes a new scheme to pretrain the neural networks, which combines the calculated persistent homology of a sublevel set filtration of images as the topology information. The topology information aims to guide the neural networks to learn non-local topological structures. To avoid the human bias in the annotated datasets, the authors utilize synthetic images generated by specific formulas during the pre-training. The authors provide empirical evaluations and show that the topology-related pre-training scheme can improve the convergence with comparable accuracy on natural image classification tasks.

**Summary Of The Review:**

Overall, this paper contributes some new ideas but the empirical evaluation is limited. Also, it's unclear to me what the model is actually learning (interpretability).

---

> ### Author Response · Authors · 2022-11-17
> **Response to the comments**
>
> Thank you very much for your time and insightful comments.
> We have revised the paper according to your suggestions.
> Below are our answers to your comments and the corresponding questions.
>
> > W1. The authors only provide empirical results, without giving theoretical analysis of the proposed pre-training scheme.
> > Q2. Using hand-crafted features to guide the training of neural networks seems alright. However, how to prove that your hand-crafted features actually reflect the global information of the training dataset? I would expect more theoretical analysis to support the main claim of this paper.
>
> Our scheme is built on well-established ideas in topology. Homology is a primary invariant to encode global characteristics of the space. For example, homology is used to classify manifolds. Manifolds are all locally R^n but globally different, which homology beautifully captures. We applied this theory in our paper for the computer vision problem.
> To count the number of connected components in a binary image, we have to look at the whole image. In this sense, the number of connected components is a global feature of a binary image. Persistent homology generalises the number of connected components (see Figure 1), and it cannot be computed locally from a portion of the image. This is the theoretical basis for why we chose persistent homology in our proposed scheme.
>
> In a different but related direction, it would be nice if we could say anything about the relationship between the computational capacity of the CNN model and the persistent homology. However, results on the (non)universal approximation for CNNs, especially those that are applicable to practical CNNs such as ResNet, are very limited. So we do not know if a CNN can theoretically approximate persistent homology or not.
>
> Empirically, global information encoded by persistent homology has been reported to add some discriminative power to conventional models (Li et al., 2014, Hofer et al. 2017, Adams et al., 2017 in the reference list). So at least, persistent homology contains certain information which is difficult to obtain by conventional techniques.
> Our paper is also in this line; we indirectly investigate the properties of learned features by looking at how the model behaves for the downstream tasks. In the following response, we explain some controlled experiments we added to support the main claim of the paper.

---

> > ### Author Response · Authors · 2022-11-17
> > **Response to the comments (continued)**
> >
> > > W2. The authors mention the weights of all layers are updated during the phase of fine tuning. However, it is unclear that the neural network is still able to preserve the learned global topological features if we update all layers’ parameters during the phase of fine tuning.
> > > Q1. It may be helpful to conduct experiments on how the learned global features change between the pre-training phase and the fine-tuning phase.
> >
> > We would like to argue as follows.
> > If the learned visual representation during the pretraining phase is totally overwritten during the finetuning phase, we should not see any improvement in the final validation accuracy compared with the one trained with the label (Label) as we see in Table 1.
> > This provides evidence that the features learned by our scheme are (at least partially) preserved during the finetuning phase, and contributes to the improved accuracy compared to the model trained only with the human-annotated labels.
> >
> > Also, we have some supporting experiments to show the effect of learned global features in the supplemental material (Section A.2).
> > Figure 6 shows the loss transition for the persistent homology regression task as in PH-PI.
> > The ImageNet pretrained model (IMN_fr) cannot be finetuned to approximate persistent homology of images if the weights of the convolutional layers are frozen. This means that the visual representation acquired through the supervised learning with ImageNet did not encode enough information necessary for the computation (approximation) of persistent homology.
> > Figure 8 shows the loss transition for the same persistent homology regression task for a different dataset. The model pretrained with our scheme (PH-PI_fr) can be finetuned to approximate persistent homology of images generated in a totally different manner than the images used for pretraining. This means that the model’s convolutional layers acquired certain features through our pretraining scheme that were relevant to approximate persistent homology in general. We also put the loss transitions for IMN and PH-PI models finetuned by updating all layers according to your question Q1. The convergence of IMN is worse than PH-PI, showing that finetuning by updating all layers does not completely override the representation learned during pretraining.
> >
> > The GradCam++ visualisation in Figure 7 also shows an interesting distinction between the ImageNet-pretrained and our models. It seems to show that IMN_fr tends to look at the edges locally while PH-PI_fr tends to look at the whole shape.
> >
> > Additionally, following the suggestion by another referee, we have added an interesting experiment in Section A.4. It compares the types of images whose labels our PH-PI predicts correctly while the ImageNet pretrained model predicts incorrectly and vice versa.
> > The results seem to support our claim that PH-PI puts more emphasis on the shape than the texture.

---

> > > ### Author Response · Authors · 2022-11-17
> > > **Response to the comments (continued)**
> > >
> > > > W3. The experiments are all conducted on small datasets of natural image classification tasks, which is not sufficient to indicate that the proposed scheme can be used in other larger datasets or different downstream tasks.
> > > > Q3. It seems that the models are only tested on rather small datasets within only the natural image classification tasks. Is it possible to conduct experiments on larger datasets on other scenarios, like segmentation or tracking, to see the performance?
> > >
> > >
> > > We added some experiments on a PH regression task in Section A.2, which were described in the response above. We know that this is not a practical task, but it directly reveals the difference between our method and the supervised training with the ImageNet classification.
> > > Also, let us explain why we have restricted ourselves to the current set of experiments, where we focused on image classification (CIFAR100, ImageNet, Animal) with varied configurations for the pretraining scheme (vectorisation, dimension, number of images, etc.).
> > > The literature (see, for example, Geirhos et al. 2019 and references therein) suggests that image classification is suitable for investigating the notions of 'local' and 'global' image features. This is why we chose image classification as the task for experiments.
> > > Our main objective is to introduce our fundamental research question: "Can a model learn anything without real data by just solving a maths problem?" and our partial answer to it: "Persistent homology can be useful".
> > > For these, we think it is effective to fix a single task and vary the parameters of the proposed learning scheme as we did in the paper.
> > > We would like to note that the images in the animal dataset contain only the silhouettes.
> > > It adds a variety to the classification problem and was indeed used to test a topological method in the literature (Hofer et al. 2017).
> > >
> > > We added an experiment (Fig. 11) of an object detection task with Faster R-CNN with ResNet50 backbones pretrained with various methods. The comparison we made may not be very meaningful as the mAP stayed low for all SSL models.
> > >
> > > It is true that we have limited computational resources available to us (RTX3090), hence not able to run massive experiments; ImageNet could be said to be “small” as of 2022, but it takes us more than 3 days to do an experiment with a single configuration (see Table 1).
> > > This research grew out of such a limited environment, and we studied in a more fundamental direction by introducing this new approach with homology. Our aim is to introduce a novel approach to visual representation learning using an idea from topology and prove that it brings a new type of image feature. We believe our experiments succeed in supporting our claim, although they are not thorough.

---

> > > > ### Comment · Reviewer_tB82 · 2022-12-12
> > > > **Thanks for the detailed response**
> > > >
> > > > Thanks very much for the detailed response! I really appreciate the efforts the authors have made and some of my concerns have been addressed. However, after reading all the reviews and rebuttals, I still feel this paper is not ready to publish. The major concerns are that 1) It's still unclear to me what the model is actually learning, and 2) The experimental validations are limited.
> > > >
> > > > I will increase my score to 5 to reflect all the above.

---

### Official Review · Reviewer_nihr · 2022-10-25

**Confidence:** 3
**Clarity, Quality, Novelty And Reproducibility:** The paper explains the problem and te…
**Correctness:** 3
**Technical Novelty And Significance:** 3
**Empirical Novelty And Significance:** 3
**Recommendation:** 6

**Strength And Weaknesses:**

Strengths: The paper explains the intentions and technical parts quite well. It is tested reasonably thorough, comparing the various vectorization techniques, as well as with other modules.

Weaknesses: it is hard to understand what is actually being learned. It would help to know what kind of images are now better classified after the pre-training and fine-tuning regiments to demonstrate the usage of structure learned in improving final task accuracy.
That is, would it be possible to show example testing images that are moving from incorrect to correct.
Also, it would be helpful to describe what types of global structures (and invariances) are eventually being learned by PH synthetic image pre-training.


**Summary Of The Paper:**

The authors propose a technique to learn local and global visual representation via Persistent Homology.
The overall structure of the proposed method is to perform self-supervised learning (regression) on synthetic images for Persistent Homology as a pre-training, and then natural images for fine-tuning.

The synthetic image utilizes frequency-based (FFT) random generation method, where there are patterns at different scales
The PH label computation (using Cubical Ripser) aims at extracting shapes using contours by using signed distance function.
The author then utilizes four different vectorization techniques: the persistence image, the persistence landscape, the Betti number curve, and the birth-life histogram.

The algorithm is tested on ImageNet and CIFAR100. It also uses 2000 binary images of animal contours for a second (mid-step) topology learning.
That is, for ImageNet for example, it uses 90 epoch synthetic data, 300 epochs for animal dataset, then 45 epochs for ImageNet.
The first experiment is testing various pre-training (including vectorization techniques) and fine-tuning training combinations, as well as other methods, such as MoCo v2 and FractralDB-pre-trained models.

The PH-pre-trained models show better convergence than without pre-training (just fine-tuning) or fine-tuned twice, and comparable to FractalDB.
Among the vectorization, all are similar, with Betti count performs worst. This may be because of the lack of persistence structure learning.
The experiments also show that the number of PH-training images is important as there is only 1600 of the animal training images.


**Summary Of The Review:**

As stated the problem and solution (technique) is explained well. However, the testing and analysis can be more in-depth.

---

> ### Author Response · Authors · 2022-11-17
> **Response to the comments**
>
> Thank you very much for your time and insightful comments.
> We have revised the paper according to your suggestions.
> Below are our answers to your comments and the corresponding questions.
>
>
> > Weaknesses: it is hard to understand what is actually being learned. It would help to know what kind of images are now better classified after the pre-training and fine-tuning regiments to demonstrate the usage of structure learned in improving final task accuracy. That is, would it be possible to show example testing images that are moving from incorrect to correct.
>
> We agree that this is a very good way to see the difference between the models,
> and it provides supportive evidence to an important question raised by all referees.
> Thank you very much for the suggestion.
>
> We added an experiment to the supplemental material in Section A.4.
> We looked at the predictions for the CIFAR100 validation dataset (100 classes x 100 images)
> of the following three models (as in Table 1):
> The model pretrained with our scheme (PH-PI),
> ImageNet-pretrained model (IMN),
> CIFAR100-pretrained model (Label).
>
> We computed the f1-score for each of the 100 classes and sorted the result in the increasing order of (f1 of PH-PI) - (f1 of IMN).
> Table 4 shows the three classes with the smallest (f1 of PH-PI) - (f1 of IMN) and three classes with the largest (f1 of PH-PI) - (f1 of IMN).
> A positive value of (f1 of PH-PI) - (f1 of IMN) > 0 means that PH-PI performed better than IMN for the class.
> (We note that PH-PI had better f1-scores than Label for all 100 classes.)
> Figure 11 shows example testing images that are moving from incorrect to correct and the other way around for the six classes appearing in Table 4.
> The result is illustrative and seems to be in agreement with our claim that our model puts more emphasis on the shape than the texture.
> We have some discussions in Section A.4.
>
>
> > Also, it would be helpful to describe what types of global structures (and invariances) are eventually being learned by PH synthetic image pre-training.
>
> The experiments in the supplemental material in Section A.2 answer the question partially.
> Figure 6 shows that the ImageNet-pretrained model failed to be fine-tuned (by updating only fully-connected layers) for approximating persistent homology. (more precisely, the PH regression task as in PH-PI).
> This means that the visual representation acquired through the supervised training with ImageNet did not encode enough information necessary for the computation (approximation) of persistent homology. This contrasts with the result in Figure 8, where PH-PI was successfully finetuned to approximate persistent homology of synthetic images that were of a different type than those used in pretraining.
> Also, GradCam++ visualisation in Figure 7 seems to show that IMN tends to look at the edges locally while PH-PI tends to look at the whole shape.

---

### Official Review · Reviewer_U7F9 · 2022-10-26

**Confidence:** 3
**Correctness:** 2
**Technical Novelty And Significance:** 2
**Empirical Novelty And Significance:** 2
**Recommendation:** 5

**Clarity, Quality, Novelty And Reproducibility:**

The paper is clearly written and easy to understand.
The idea is simple and easy to follow. I think it is novel and interesting, but not certain whether it is effective for other datasets and other model architectures.
The authors will release the code to ensure reproducibility.

**Details Of Ethics Concerns:**

No ethics concerns.

**Strength And Weaknesses:**

Strength:
- Training a better representation without using real data is an interesting direction. This paper presents an innovative way to learn holistic features.

Negative:
- The effectiveness of this method is uncertain. It is uncertain whether the feature learned using this synthetic data can work well together with real data and tasks. The authors mentioned, "The image features obtained by PH are topological and global, whereas the features extracted through iterated convolutions by CNNs are mostly local."  This disadvantage might not hold for various cases. Many models can encode long-term context effectively.
- The experiment is not persuasive enough to show whether this method can be generalized to in-the-wild real data. The binary image is an effective way to prove the concept, but it is also helpful to include results for more challenging real data to show its effectiveness.


**Summary Of The Paper:**

This paper proposes to use synthetic data to learn a better representation that can enforce the model to learn holistic features. The synthetic data is generated using persistent homology (PH). This method does not require real images. In the experiment, authors try to apply the learnt representation to downstream tasks. The experiment is done on the animal dataset, which consist of binary images.

**Summary Of The Review:**

The paper proposes an interesting way to generate synthetic data. Then use this data to train a better representation that can encode holistic information. The experiment is limited to a binary dataset, which makes it hard to judge whether the training strategy can work well on real-world data. Due to the challenge and complexity of real data, many existing models can already encode long-term context and have large receptive fields. This fact makes the advantage of the proposed method seem less significant. Based on these reasons, I think the paper is below the threshold of ICLR.

---

> ### Author Response · Authors · 2022-11-17
> **Response to the comments**
>
> Thank you very much for your time and insightful comments.
> We have revised the paper according to your suggestions.
> Below are our answers to your comments and questions.
>
> > The experiment is limited to a binary dataset, which makes it hard to judge whether the training strategy can work well on real-world data.
>
> Our experiments were performed on CIFAR100 and ImageNet in addition to a binary dataset (Animal). Please look at Section 3.3, especially Table 1, if you have missed it.
> C100 indicates CIFAR100, IMN-1k indicates ImageNet-1k, and ANM indicates the animal dataset.
> (In the initial version of the paper, IMN was used to indicate both the dataset and the model.
> We have revised to write IMN-1k to indicate the dataset. We are sorry for the confusing notation.)
>
> Our PH-PI does not surpass the supervised model (IMN), but it is comparable with FractalDB-10k (FDB-10k). FDB-10k is also pretrained with synthetic images, but it takes much longer to pretrain.
>
> To show the convergence rate for a larger dataset, we have added Figure 9, showing the transition of training and validation accuracies during finetuning for ImageNet-1k. The tendency is quite similar to that for CIFAR100 (Figure 4). The IMN surpasses all other models in both training and validation. The convergence behaviours of FDB-10k and our PH-PI are similar, sitting between IMN and Scratch.
>
>
> > many existing models can already encode long-term context and have large receptive fields. This fact makes the advantage of the proposed method seem less significant.
>
> We added an experiment with a larger CNN architecture, ResNet101. Figure 10 shows the transition of training and validation accuracies during finetuning for CIFAR100. The result is quite similar to that of ResNet50.
>
> Combined with Figure 9 explained above, our results seem to be stable with respect to the size of the dataset and the CNN model to some extent.
>
> Furthermore, we would like to note the advantage of the proposed method with a relatively lightweight model architecture of ResNet50.
> Model architecture and learning schemes are two different approaches to improved visual representation learning. Even if the gain observed with a lightweight model would be superseded by the improvement in the model architecture, it still suggests that the learning scheme offers something that helps learning. Our pretraining scheme is effectively helping the model to acquire long-term context without changing its inborn capacity associated with the architecture.
>
>
> > The authors mentioned, "The image features obtained by PH are topological and global, whereas the features extracted through iterated convolutions by CNNs are mostly local." This disadvantage might not hold for various cases. Many models can encode long-term context effectively.
>
> We agree that our claim was not accurate.
> We changed the phrasing as follows:
> --
> The image features obtained by PH are topological and global,
> whereas the features acquired by CNNs through supervised learning with ImageNet are often biased towards textures.
> —
>
> We believe "locality" is one of the fundamental limitations of convolution (with limited kernel size and receptive field), as evidenced in Geirhos et al. 2019 and references therein.
> We have some experiments in this direction in the supplementary material Section A.2.
> Figure 6 shows that the ImageNet pretrained ResNet50 failed to be finetuned (by updating only fully-connected layers) for approximating persistent homology (more precisely, the PH regression task as in PH-PI).
> This means that the visual representation acquired through the supervised training with ImageNet did not encode enough information necessary for the computation (approximation) of persistent homology. This contrasts with the result in Figure 8, where PH-PI was finetuned to approximate persistent homology of synthetic images that were of a different type than those used in pretraining.
> Also, GradCam++ visualisation in Figure 7 seems to show that IMN tends to look at the edges locally while PH-PI tends to look at the whole shape.
>
> Following the suggestion by another referee, we have also added an interesting experiment in Section A.4. It compares the types of images whose labels our PH-PI predicts correctly while the ImageNet pretrained model predicts incorrectly and vice versa.
> The results seem to support our claim that PH-PI puts more emphasis on the shape than the texture.

---

> > ### Author Response · Authors · 2022-11-17
> > **Response to the comments (continued)**
> >
> > > The effectiveness of this method is uncertain. It is uncertain whether the feature learned using this synthetic data can work well together with real data and tasks.
> >
> > We agree that one of the most important goals of SSL is better performance in practical tasks.
> > However, we believe that our scheme provides an interesting approach to a fundamental question in learning
> > by asking “can a model learn anything without real data by just solving a maths problem?"
> > Our proposed method is a proof-of-concept for incorporating mathematical knowledge into data-driven learning. Our method achieves a comparable performance (on the validation accuracy of ImageNet-1k and CIFAR100) with the previous pretraining method with totally synthetic images (FractalDB). This direction of study has just begun, and it still has many miles to go before achieving comparable performance with supervised or self-supervised learning methods which use natural images.

---

### Decision · Program_Chairs · 2023-01-20

**Decision:**

Reject

**Justification For Why Not Higher Score:**

See the weakness and summary of AC-reviewer meeting. The flaw is critical from the perspective of reviewers/meta-reviewer. Therefore, we recommend to reject the work.

**Justification For Why Not Lower Score:**

N/A

**Metareview: Summary, Strengths And Weaknesses:**

The paper proposes to improve the ability of CNNs to capture long-term dependencies by supervising with an analytical topological signature on randomly generated synthetic images. The idea is novel, and the method is tested on a few popular datasets. The paper received mixed review scores (5, 5, 6). The major issues include the interpretation to how CNNs learn long-term dependency is inadequate and how this method differs from popular approaches such as transformers in learning global context is not well articulated/evaluated.

**Summary Of Ac-Reviewer Meeting:**

The major issues are: 1) the interpretation to how CNNs learn long-term dependency is inadequate; and 2) how this method differs from popular approaches such as transformers in learning global context is not well articulated/evaluated.

While the authors provided a rebuttal, the reviewers are still not fully convinced. In particular, because vision transformers have been very popular in the community and a major claim is that the design captures global context, this comparison (at theoretical and/or at experiment level) is needed.

Therefore, the committee recommends to reject the submission.